# Investigation of the Anti-Methicillin-Resistant *Staphylococcus aureus* Activity of (+)-Tanikolide- and (+)-Malyngolide-Based Analogues Prepared by Asymmetric Synthesis

**DOI:** 10.3390/ijms22126400

**Published:** 2021-06-15

**Authors:** Joseph Breheny, Cian Kingston, Robert Doran, Joao Anes, Marta Martins, Séamus Fanning, Patrick J. Guiry

**Affiliations:** 1Centre for Synthesis and Chemical Biology, School of Chemistry, University College Dublin, Belfield, Dublin D04 N2E5, Ireland; joseph.breheny@ucdconnect.ie (J.B.); robert.doran@ucdconnect.ie (R.D.); 2Synthesis and Solid State Pharmaceutical Centre, School of Chemistry, University College Dublin, Belfield, Dublin D04 N2E5, Ireland; cian.kingston@ucdconnect.ie; 3UCD-Centre for Food Safety, University College Dublin, Belfield, Dublin D04 N2E5, Ireland; joao.anes@ucdconnect.ie (J.A.); marta.martins@ucd.ie (M.M.); sfanning@ucd.ie (S.F.)

**Keywords:** (+)-tanikolide, (−)-malyngolide, asymmetric synthesis, anti-methicillin-resistant *Staphylococcus aureus* activity

## Abstract

Herein, we report antibacterial and antifungal evaluation of a series of previously prepared (+)-tanikolide analogues. One analogue, (4*S*,6*S*)-4-methyltanikolide, displayed promising anti-methicillin-resistant *Staphylococcus aureus* activity with a MIC of 12.5 µg/mL. Based on the antimicrobial properties of the structurally related (−)-malyngolide, two further analogues (4*S*,6*S*)-4-methylmalyngolide and (4*R*,6*S*)-4-methylmalyngolide bearing a shortened *n*-nonyl alkyl side chain were prepared in the present study using a ZrCl_4_-catalysed deprotection/cyclisation as the key step in their asymmetric synthesis. When these were tested for activity against anti-methicillin-resistant *Staphylococcus aureus*, the MIC increased to 50 µg/mL.

## 1. Introduction

We developed a ZrCl_4_-catalysed one-pot deprotection/cyclisation synthetic protocol for the construction of δ-lactones [1]. The methodology was subsequently applied in the asymmetric synthesis of both enantiomers of a mosquito attractant pheromone [2], substituted tetrahydropyrans which provided useful synthons for the enantioselective synthesis of (+)-*exo*- and (+)-*endo*-brevicomin [3] and for the efficient synthesis of (−)-frontalin and (−)-*exo*-isobrevicomin [4]. Finally, of relevance to this report, the methodology was applied to the asymmetric synthesis of (+)-tanikolide, **1**, affording the δ-lactone based natural product in an overall yield of 26.4% [5]. (+)-Tanikolide **1** displays strong toxicity against brine shrimp and snails and interesting antifungal activity against *C. albicans* [6]. *C. albicans* is the most common fungal pathogen of human diseases and together with other *Candida* species are responsible for ca. 400,000 life-threatening infections per annum with a mortality rate as high as 40% [7,8]. Current therapeutic drugs for *Candida* infections include members of five classes of compounds: polyenes, allylamines, azoles, fluoropyrimidines and echinocandins [9] with amphotericin B, terbinafine, fluconazole, 5-fluorocytosine and caspofungin being the most well-known examples [10].

(+)-Tanikolide **1** is structurally closely related to the marine antibiotic (−)-malyngolide, **2**, with three key differences illustrated in Figure 1; a shortened alkyl side chain (Figure 1, **2** difference **A**), opposite configuration at the quaternary stereocentre (Figure 1, **2** difference **B**) and a methyl group α- to the carbonyl (Figure 1, **2** difference **C**). Interestingly, despite the similarity to (+)-tanikolide **1**, (−)-malyngolide **2** displays no activity against *C. albicans* [6]. However, (−)-malyngolide **2** does display anti-microbial activity against *Myobacterium smegmatis*, *Staphylococcus aureus*, *Bacillus subtilis* and *Streptococcus pyogenes* [11]. The bacterium *Staphylococcus aureus* is among one of the most aggressive human pathogenic agents [12]. Antibiotic resistance to *S. aureus* is a major medical issue [13] and is the result of the widespread use of antibacterial antibiotics since the 1940s [14]. The most effective antibiotics for MRSA eradication are vancomycin, linezolid and a few others in combination with vancomycin. Daptomycin, clindamycin, doxycycline, tigecyclin and trimethoprim–sulfamethoxazole combination is also efficient against most MRSA strains [15]. The search for new compounds to act as antifungal and antimicrobial agents is an active field of research and, herein, we report our results with analogues of (+)-tanikolide **1** and (−)-malyngolide **2**.

In addition, to our reported synthesis of (+)-tanikolide **1**, we wished to probe the biological importance of the position of the methyl group and hence the four β-methyl modified analogues (**3**–**6**) were synthesised using the same δ-lactone forming methodology with the aim to enhance the antifungal activity against *C. albicans* (Figure 2) [5]. These analogues (**3**–**6**) were subsequently biologically evaluated, the results of which we report now (Table 1).

## 2. Results

The four β-methyl (+)-tanikolide based analogues (**3**-**6**) were submitted for biological testing to ascertain if they exhibited any antifungal and antimicrobial activity. The compounds were tested against *Candida albicans* and *Candida parapsilosis*. Unfortunately, the compounds displayed no inhibition of growth even at concentrations as high as 800 µg/mL. However, the series of compounds were also tested for activity against Gram-positive and Gram-negative bacterial strains, methicillin-resistant *Staphylococcus aureus* (MRSA) and *Escherichia coli* (*E. coli*) (Table 1). Although the compounds showed no activity against *E. coli*, analogue **5** was found to exhibit promising results against MRSA with an MIC of 12.5 µg/mL. This compares favourably with the typical MIC values of vancomycin (4.8 µg/mL) [16] and linezolid (0.1–4 mg/L) [17]. Analogue **5** was shown to be stable for the duration of the assay. The configuration of a methyl group—to the carbonyl had a dramatic effect upon the specific activity of the compound, as shown by analogue **6** which displayed no bioactivity. Analogue **5** bears the opposite stereochemistry at the quaternary carbon centre to **1**, upon which the analogues were initially designed. Interestingly, the configuration is the same as found in (−)-malyngolide **2**, a known anti-microbial agent.

In an effort to further increase the efficacy of these potential anti-MRSA agents, we wished to synthesise analogues bearing the shortened *n*-nonyl side chain found in (−)-malyngolide **2**. The optimal stereochemistry of the β-methyl group will be determined once again by the synthesis and evaluation of both diastereomers. A number of approaches to asymmetric synthesis of **2** have been published since the first report by Mukaiyama in 1980 [18], including the use of chiral auxiliary [19,20,21,22,23,24,25,26,27], chiral pool [28,29,30,31,32,33,34,35,36,37], other asymmetric syntheses [38,39,40] and catalytic asymmetric syntheses [41,42,43,44,45,46,47,48,49].

The synthesis of (−)-malyngolide based analogues **7** and **8** was adapted from our initial synthesis of (+)-tanikolide based compounds **3**–**6** (Figure 3) [5]. The first step was monoalkylation of phosphonate **9** which afforded intermediate **10** in a yield of 51%. A Horner-Wadsworth-Emmons reaction with 35% aqueous formaldehyde successfully gave the desired terminal alkene of intermediate **11** in a yield of 73%. DIBAL reduction of the ethyl ester provided allylic alcohol **12** in 25% yield. A Sharpless asymmetric epoxidation using Ti(OiPr)_4_, (−)-diisopropyltartrate and cumene hydroperoxide was used to afford intermediate **13** with the desired stereochemistry in a yield of 80%. The *ee* was subsequently determined after benzyl protection of the primary alcohol in **13** due to the absence of a chromophore on the unprotected epoxide. The stereochemistry of the product was assigned based on extensive NOE experiments carried out on analogues **3**–**6** [5]. Protection of the cohol was achieved using sodium hydride as a base with benzyl bromide in the presence of tetrabutylammonium iodide to give **14** in a yield of 86% with an *ee* greater than 99% (see Appendix A for reference chiral SFC chromatograms). At this point a diol protection/bromination of crotonaldehyde **16** was carried out which provided intermediate **15** in 83% yield. Intermediate **15** was then applied in a copper-catalysed Grignard addition to epoxide **14** which, upon separation via silica gel column chromatography, provided diastereomers **18** and **21** in an overall yield of 69% [50]. **18** was subjected to our developed ZrCl_4_-catalysed one-pot deprotection/cyclisation technique to afford diastereomeric acetals **19** and **20** in a yield of 92%. Conversion to the desired δ-lactone **21** was achieved using the Lewis acid BF_3_.OEt_2_ and mCPBA with a yield of 52% [51,52]. Hydrogenolysis of the benzyl ether was carried out using Pearlman’s catalyst at 25 bar pressure to provide (4*S*, 6*S*)-4-methyl-malyngolide **7** in a yield of 94%. Diastereomer **21** was subjected to a similar synthetic sequence to afford (4*R*, 6*S*)-4-methyl-malyngolide **8** with yields of 89, 42 and 65% obtained for the cyclisation, oxidation and deprotection steps, respectively.

With the synthesis complete, the new analogues **7** and **8** were tested for their biological activity (Table 2). The results indicate that the *n*-nonyl chain had a significant deleterious effect on the anti-MRSA action of the compounds. Analogues **7** and **8** displayed similar activity with their lowest MIC and MBC values of 50 µg/mL. Further synthesis of modified analogues is currently underway in an effort to enhance the biological activity of this interesting class of compounds.

## 3. Conclusions

In summary, we have determined anti-methicillin-resistant *Staphylococcus aureus* activity (MIC of 12.5 µg/mL) by a novel β-methyl analogue **5** of (+)-tanikolide **1**. In an effort to improve upon this activity, two further analogues **7** and **8** bearing a shortened *n*-nonyl alkyl side chain were prepared in the present study using a ZrCl_4_-catalysed deprotection/cyclisation as the key step. When these were tested for activity against anti-methicillin-resistant *Staphylococcus aureus* the MIC increased to 50 µg/mL. It is hoped the results described above will lead to further improvements in this class of potentially potent anti-methicillin-resistant *Staphylococcus aureus* compounds.

## 4. Materials and Methods—Chemistry

Unless otherwise noted, reactions were performed with rigorous exclusion of air and moisture, under an inert atmosphere of nitrogen in flame-dried glassware with magnetic stirring using anhydrous solvents. N_2_-flushed stainless steel cannulas or plastic syringes were used to transfer air and moisture-sensitive reagents. All reagents were obtained from commercial sources and used without further purification unless otherwise stated. All anhydrous solvents were obtained from commercial sources and used as received with the following exceptions: diethyl ether (Et_2_O), dichloromethane (CH_2_Cl_2_) and toluene (PhCH_3_) were dried by passing through activated alumina columns. Powdered activated 4 Å molecular sieves were purchased from Sigma Aldrich and were stored in an oven at 120 °C. In vacuo refers to the evaporation of solvent under reduced pressure on a rotary evaporator. Thin-layer chromatography (TLC) was performed on aluminium plates pre-coated with silica gel F254. They were visualised with UV-light (254 nm) fluorescence quenching, or by charring with Hanessian’s staining solution (cerium molybdate, H_2_SO_4_ in water), basic potassium permanganate staining solution (potassium permanganate, K_2_CO_3_ and NaOH in water), or an acidic vanillin staining solution (vanillin, H_2_SO_4_ in ethanol). Flash column chromatography was carried out using 40–63 μm, 230–400 mesh silica gel.

^1^H NMR spectra were recorded on a 300, 400 or 500 MHz spectrometer. ^13^C NMR spectra were recorded on a 400 or 500 MHz spectrometer at 101 or 126 MHz. ^19^F NMR spectra were recorded on a 400 MHz spectrometer at 376 MHz. Chemical shifts (δ) are reported in parts per million (ppm) downfield from tetramethylsilane and for ^1^H NMR are referenced to residual proton in the NMR solvent (CDCl_3_ = δ 7.26 ppm). ^13^C NMR are referenced to the residual solvent peak (CDCl_3_ = δ 77.16 ppm). All ^13^C spectra are ^1^H decoupled. NMR data are represented as follows: chemical shift (δ ppm), integration, multiplicity (s = singlet, d = doublet, t = triplet, q = quartet, dd = double doublet, m = multiplet, app. d = apparent doublet, app. t. = apparent triplet), coupling constant (*J*) in Hertz (Hz). High resolution mass spectra [electrospray ionisation (ESI-TOF)] (HRMS) were measured on a micromass LCT orthogonal time-of-flight mass spectrometer with leucine enkephalin (Tyr-Gly-Phe-Leu) as an internal lock mass. Infrared spectra were recorded on a FT-IR spectrometer and are reported in terms of wavenumbers (ν_max_) with units of reciprocal centimetres (cm^−1^). Microwave experiments were conducted in a CEM Discover S-class microwave reactor with controlled irradiation at 2.45 GHz using standard microwave process Pyrex vials. Reaction time reflects time at the set reaction temperature maintained by cycling of irradiation (fixed hold times). Optical rotation (α) values were measured at room temperature and specific rotation ([α]_D_^20^) values are given in deg.dm^−1^.cm^3^.g^−1^. Melting points were determined in open capillary tubes. Supercritical fluid chromatography (SFC) was performed on a Waters UPC^2^ system using a Chiralpak IB column.

### 4.1. Ethyl 2-(diethoxyphosphoryl)undecanoate (**10**)

NaH (60% in mineral oil, 6.0 g, 150 mmol) was placed in a dry 500 mL two-necked room-bottom flask (RBF) containing a magnetic stirrer bar under an inert atmosphere, was washed with anhydrous hexanes (2 × 20 mL) and dried under high vacuum. Dry THF (250 mL) was added to the reaction flask and triethylphosphonoacetate **9** (19.8 mL, 100 mmol) in dry THF (30 mL) was added dropwise over 20 min to the reaction mixture, with evolution of H_2_ gas. NaI (3.7 g, 25 mmol) was added to the reaction flask followed by dropwise addition of 1-bromononane (9.6 mL, 50 mmol) and the reaction mixture was heated at reflux for 24 h. The reaction mixture was quenched with H_2_O (100 mL) and the aqueous layer was extracted with ether (3 × 100 mL). The combined organic layers were washed with H_2_O (100 mL) and brine (100 mL) and dried with anhydrous Na_2_SO_4_. The solvent was removed in vacuo and the crude product was purified by silica gel column chromatography (pentane/ether, 9:1 → 4:1) to yield **10** as a colourless oil (8.93 g, 51%).

Spectroscopic analysis of **10**: R_f_ = 0.20 (pentane/diethyl ether, 1:9); IR (neat): ν_max_ = 3477, 2926, 2854, 1729, 1465, 1250, 1029 cm^−1^; ^1^H NMR (400 MHz, CDCl_3_): *δ* 4.28–4.09 (m, 6 H), 2.92 (ddd, *J* = 22.5, 11.1, 3.7 Hz, 1 H), 2.04–1.90 (m, 1 H), 1.90–1.77 (m, 1 H), 1.50–1.09 (m, 23 H), 0.88 (t, *J* = 6.9 Hz, 3 H) ppm; ^13^C NMR (126 MHz, CDCl_3_): *δ* 169.2 (d, *J* = 4.8 Hz), 62.6 (d, *J* = 6.6 Hz), 62.5 (d, *J* = 6.6 Hz), 61.2, 45.8 (d, *J* = 131.0 Hz), 31.8, 29.4, 29.2 (d, *J* = 4.6 Hz), 29.0, 28.4, 28.3, 26.9 (d, *J* = 5.0 Hz), 22.6, 16.3 (d, *J* = 4.0 Hz), 16.3 (d, *J* = 4.0 Hz), 14.1, 14.0 ppm; ^31^P NMR (162 MHz, CDCl_3_) *δ* 22.98 ppm; HRMS (ESI-TOF): calcd. for C_17_H_35_O_5_PNa [M + Na]^+^ 373.2120; found 373.2108. (see Appendix A for ^1^H and ^13^C NMR spectra).

### 4.2. Ethyl 2-methyleneundecanoate (**11**)

Phosphate ester **10** (8.93 g, 25.5 mmol) was placed in a 250 mL two-necked RBF containing a magnetic stirrer bar, followed by deionised water (30 mL), K_2_CO_3_ (14.1 g, 101.9 mmol) and aqueous formaldehyde (16.5 mL, 37%, 203.8 mmol). The reaction mixture was stirred at 85 °C for 18 h. The reaction mixture was extracted with diethyl ether (3 × 100 mL). The combined organic layers were washed with H_2_O (100 mL) and brine (100 mL) and dried with anhydrous Na_2_SO_4_. Excess solvent was removed in vacuo and the crude product was purified by silica gel column chromatography (pentane/diethyl ether, 9:1) to yield **11** as a colourless oil (4.24 g, 73%). (see Appendix A for ^1^H and ^13^C NMR spectra).

Spectroscopic analysis of **11**: R_f_ = 0.70 (pentane/diethyl ether, 9:1); IR (neat): ν_max_ = 2926, 2856, 1720, 1179, 1147 cm^−1^; ^1^H NMR (400 MHz, CDCl_3_) *δ* 6.12 (d, *J* = 1.5 Hz, 1 H), 5.50 (d, *J* = 1.5 Hz, 1 H)**,** 4.20 (q, *J* = 7.1 Hz, 2 H), 2.29 (t, *J* = 7.7 Hz, 2 H), 1.50–1.41 (m, 2 H), 1.37–1.19 (m, 15 H), 0.88 (t, *J* = 7.0 Hz, 3 H) ppm; ^13^C NMR (126 MHz, CDCl_3_) *δ* 167.4, 141.2, 124.0, 60.5, 31.9, 31.8, 29.5, 29.4, 29.3, 29.2, 28.4, 22.7, 14.2, 14.1 ppm; HRMS (ESI-TOF): calcd. for C_14_H_26_O_2_Na [M + Na]^+^ 249.1831; found 249.1840.

### 4.3. 2-Methyleneundecan-1-ol (**12**)

Allylic ester **11** (4.24 g, 18.71 mmol) was placed in a dry 100 mL two-necked RBF containing a magnetic stirrer bar and dissolved in dry THF (55 mL), under an inert atmosphere. The reaction mixture was cooled to −30 °C and DIBAL (25 wt.% in toluene, 9.5 mL, 41 mmol) was added dropwise over 40 min and the reaction mixture was stirred for 1 h. The reaction mixture was quenched with diethyl ether (5 mL) and a saturated solution of Rochelle’s salt (potassium sodium tartrate) (50 mL). The reaction mixture was stirred for 16 h at room temperature. The product was extracted with diethyl ether (3 × 100 mL). The combined organic layers were washed with H_2_O (100 mL) and brine (100 mL) and dried with anhydrous Na_2_SO_4_. Excess solvent was removed in vacuo and the crude product was purified by silica gel column chromatography (pentane/diethyl ether, 4:1) to yield **12** as a colourless oil (0.856 g, 25%). (see Appendix A for ^1^H and ^13^C NMR spectra).

Spectroscopic analysis of **12**: R_f_ = 0.19 (pentane/diethyl ether, 4:1); IR (neat): ν_max_ = 3323, 2926, 2856, 1653, 1465, 1027 cm^−1^; ^1^H NMR (400 MHz, CDCl_3_) *δ* 5.03–4.98 (m, 1 H), 4.88–4.83 (m, 1 H), 4.07 (s, 2 H), 2.05 (t, *J* = 7.6 Hz, 2 H), 1.50–1.37 (m, 2 H), 1.36–1.17 (m, 12 H), 0.88 (t, *J* = 6.9 Hz, 3 H) ppm; ^13^C NMR (126 MHz, CDCl_3_) *δ* 148.3, 107.9, 64.9, 32.0, 30.9, 28.6, 28.5, 28.4, 28.3, 26.8, 21.7, 13.1 ppm; HRMS (EI-TOF): calcd. for C_12_H_24_O [M]^+^ 184.1828; found 184.1827.

### 4.4. (R)-(2-Nonyloxiran-2-yl)methanol (**13**)

Molecular sieves (4 Å, 400 mg) and dry CH_2_Cl_2_ (11.5 mL) were added to a dry 50 mL Schlenk tube containing a magnetic stirrer bar, followed by Ti(O*^i^*Pr)_4_ (0.141 mL, 0.464 mmol) and (−)-diisopropyltartrate (0.146 mL, 0.697 mmol), at −35 °C under an inert atmosphere. The reaction mixture was stirred for 30 min. Allylic alcohol **12** (0.856 g, 4.64 mmol) was added and the mixture was stirred for 30 min. Cumene hydroperoxide (1.37 mL, 9.29 mmol) was added over 20 min. The reaction temperature was increased to −25 °C and the progress of the reaction was monitored by TLC until the consumption of the alcohol. Upon reaction completion at 18 h, the reaction mixture was quenched with saturated sodium bicarbonate solution (1 mL) and ether (5 mL) and the resulting mixture was stirred for 2 h at room temperature. The reaction mixture was filtered through a pad of Celite^®^ and concentrated in vacuo. The epoxide was purified by silica gel column chromatography (pentane/diethyl ether, 9:1 → 4:1) to yield epoxide **13** as a colourless oil (0.743 g, 80%, > 99% *ee*). (The *ee* was calculated by SFC analysis of benzyl-protected epoxide **7** (Waters Acquity UPC^2^, Chiracel IB, scCO_2_/isopropanol = 95:5, flow rate = 2 mL min^−1^)). (see Appendix A for ^1^H and ^13^C NMR spectra).

Spectroscopic analysis of **13**: R_f_ = 0.22 (pentane/diethyl ether, 3:2); SFC: R_t_ (*R)* = 1.543 min (major); R_t_ (*S*) = 2.215 min (minor); [α]_D_^20^ = + 6.3 (c = 1.0, CHCl_3_); IR (neat): ν_max_ = 3430, 2926, 2856, 1466, 1047 cm^−1^; ^1^H NMR (400 MHz, CDCl_3_) *δ* 3.75 (dd, *J* = 12.3, 4.4 Hz, 1 H), 3.61 (dd, *J* = 12.3, 8.6 Hz, 1 H), 2.86 (d, *J* = 4.7 Hz, 1 H), 2.64 (d, *J* = 4.7 Hz, 1 H), 1.83–1.66 (m, 2 H), 1.48 (dt, *J* = 14.0, 7.5 Hz, 1 H), 1.40–1.14 (m, 14 H), 0.85 (t, *J* = 6.8 Hz, 3 H) ppm; ^13^C NMR (101 MHz, CDCl_3_) *δ* 62.7, 59.8, 49.8, 32.0, 31.8, 29.7, 29.4, 29.2, 24.6, 22.6, 14.1 ppm; HRMS (ESI-TOF): calcd. for C_12_H_24_O_2_Na [M + Na]^+^ 223.1674; found 223.1683.

### 4.5. (S)-2-[Benzyloxy)methyl]-2-undecyloxirane (**14**)

NaH (60% in mineral oil, 0.175 g, 4.379 mmol) was placed in a dry 100 mL two-necked RBF containing a magnetic stirrer bar under an inert nitrogen atmosphere, washed with anhydrous hexanes (2 × 5 mL) and dried under high vacuum. Dry THF (14.6 mL) was added and the reaction vessel cooled to 0 °C. Epoxide **13** (0.731 g, 3.649 mmol) was dissolved in dry THF (2 mL) and added to the reaction mixture, which was stirred for 30 min. Benzyl bromide (0.46 mL, 3.83 mmol) was added dropwise followed by tetra-*n*-butylammonium iodide (0.674 g, 1.825 mmol). The reaction mixture was stirred at 0 °C for 30 min and then at room temperature for 1 h. The reaction mixture was quenched with H_2_O (10 mL) and the aqueous layer extracted with diethyl ether (3 × 15 mL). The organic layers were combined and washed with H_2_O (25 mL) and brine (25 mL) and dried with anhydrous Na_2_SO_4_. The solvent was removed in vacuo and the crude product purified by silica gel column chromatography (pentane/diethyl ether, 9:1) to yield **14** as a colourless oil (0.911 g, 86%). (see Appendix A for ^1^H and ^13^C NMR spectra).

Spectroscopic analysis of **14**: R_f_ = 0.60 (pentane/diethyl ether, 4:1); [α]_D_^20^ = −3.4 (c = 1.0, CHCl_3_); IR (neat): ν_max_ = 2926, 2854, 1454, 1217, 1095 cm^−1^; ^1^H NMR (500 MHz, CDCl_3_) *δ* 7.39–7.26 (m, 5 H), 4.59 (d, *J* = 12.0 Hz, 1 H), 4.54 (d, *J* = 12.0 Hz, 1 H), 3.61 (d, *J* = 11.1 Hz, 1 H), 3.47 (d, *J* = 11.1 Hz, 1 H), 2.71 (d, *J* = 4.8 Hz, 1 H), 2.64 (d, *J* = 4.8 Hz, 1 H), 1.87–1.75 (m, 1 H), 1.61–1.50 (m, 1 H), 1.42–1.17 (m, 14 H), 0.88 (t, *J* = 7.0 Hz, 3 H) ppm; ^13^C NMR (101 MHz, CDCl_3_) *δ* 138.0, 128.3, 127.7, 127.6, 73.2, 71.9, 58.6, 50.3, 32.0, 31.9, 29.7, 29.49, 29.5, 29.3, 24.6, 22.6, 14.1 ppm; HRMS (ESI-TOF): calcd. for C_19_H_30_O_2_Na [M + Na]^+^ 313.2144; found 313.2153.

### 4.6. 2-(2-Bromopropyl)-1,3-dioxane (**15**)

Anhydrous acetonitrile (50 mL) was added to a 250 mL RBF containing a magnetic stirrer bar under an inert nitrogen atmosphere and cooled to 0 °C. Crotonaldehyde (**16**) (4.1 mL, 50 mmol) was added followed by dropwise addition of TMSBr (7.9 mL, 60 mmol) and the reaction mixture was stirred for 5 min prior to the dropwise addition of propan-1,3-diol (**17**) (4.3 mL, 60 mmol). The reaction mixture was stirred for 2.5 h at 0 °C, the warmed to room temperature and quenched into a solution of pentane (150 mL) and Na_2_CO_3_ (50 mL, 10% *w*/*v*). The solution was stirred for 5 min and added to a separating funnel. Three layers were observed, the top layer containing pentane and the product, the middle layer containing acetonitrile and the product and the bottom aqueous layer. The aqueous layer was run-off and extracted with pentane (10 mL) and sodium thiosulfate (50 mL, 10% *w*/*v*). The organic fractions were combined, washed with water (3 × 60 mL) and dried with anhydrous Na_2_SO_4_. Excess solvent was removed in vacuo and the remaining yellow solution was purified by high-vacuum distillation (bath temperature 105 °C, neck temperature 72 °C) to yield **15** as a colourless oil (8.66 g, 83%). (see Appendix A for ^1^H and ^13^C NMR spectra).

Spectroscopic analysis of **15**: R_f_ = 0.38 (pentane/diethyl ether, 9:1); IR (neat): ν_max_ = 2964, 2856, 1379, 1140 cm^−1^; ^1^H NMR (400 MHz, CDCl_3_) *δ* 4.70 (dd, *J* = 7.1, 3.4 Hz, 1 H), 4.32–4.14 (m, 1 H), 4.13–3.98 (m, 2 H), 3.85–3.64 (m, 2 H), 2.22–1.86 (m, 3 H), 1.68 (d, *J* = 6.8 Hz, 3 H), 1.38–1.24 (m, 1 H) ppm; ^13^C NMR (101 MHz, CDCl_3_) *δ* 100.5, 66.7, 46.0, 45.6, 26.7, 25.7 ppm; HRMS (EI-TOF): calcd. for C_7_H_12_O_2_^79^Br [M−H]^+^ 207.0021 and C_7_H_12_O_2_^81^Br [M−H]^+^ 209.0000; found 207.0023 and 208.9995, respectively. All physical data was identical to those previously reported [5].

### 4.7. (2S,4S)-4-((Benzyloxy)methyl)-1-(1,3-dioxan-2-yl)-2-methyltridecan-4-ol ((2S,4S)-**18**) & (2R,4S)-4-((benzyloxy)methyl)-1-(1,3-dioxan-2-yl)-2-methyltridecan-4-ol ((2R,4S)-**22**)

The Grignard reagent was prepared by addition of bromide **15** (1.941 g, 9.285 mmol) to a dry 25 mL two-necked RBF containing a magnetic stirrer bar, magnesium turnings (0.226 mg, 9.285 mmol) and a crystal of I_2_ in anhydrous THF (9 mL) under an inert nitrogen atmosphere followed by heating to reflux for 1.5 h min. The solution was cooled to room temperature then transferred by cannula to a dry 25 mL two-necked RBF containing copper (I) iodide (0.059 g, 0.310 mmol) at −45 °C and stirred for 30 min. Benzyl epoxide **14** (0.899 g, 3.095 mmol) in anhydrous THF (3 mL) was added dropwise over 20 min and stirring was continued for a further 2 h at −45 °C. The reaction was quenched by the addition of solid NH_4_Cl (0.90 g) and saturated NH_4_Cl solution (5 mL) and the solution was stirred at room temperature for 10 min. The solution was extracted with ethyl acetate (6 × 30 mL) and the combined organic layers were washed with water (50 mL) and brine (50 mL) and dried with anhydrous Na_2_SO_4_. The solvent was removed in vacuo and the crude product purified by silica gel column chromatography (pentane/dichloromethane/ether, 5.5:3:1.5, repeated three times) to yield (2*S*,4*S*)-**18** as a colourless oil (0.397 g, 30%), (2*R*,4*S*)-**22** as a colourless oil (0.424 g, 33%) and a mixture (0.072 g, 6%). (see Appendix A for ^1^H and ^13^C NMR spectra of compound **8** and Appendix A for ^1^H and ^13^C NMR spectra of compound **22**).

Spectroscopic analysis of (2*S*,4*S*)-**18**: R_f_ = 0.38 (pentane/diethyl ether, 1:1); [α]_D_^20^ = −3.5 (c = 0.7, CHCl_3_); IR (neat): ν_max_ = 3446, 2962, 2852, 1454, 1261, 1088 cm^−1^; ^1^H NMR (500 MHz, CDCl_3_) *δ* 7.38–7.24 (m, 5 H), 4.62–4.47 (m, 3 H), 4.08 (dd, *J* = 12.1, 4.9 Hz, 2 H), 3.72 (td, *J* = 12.1, 2.4 Hz, 2 H), 3.34 (d, *J* = 8.9 Hz, 1 H), 3.30 (d, *J* = 8.9 Hz, 1 H), 2.45 (s, 1 H), 2.13–1.99 (m, 1 H), 1.87 (dtd, *J* = 13.5, 6.8, 4.6 Hz, 1 H), 1.68–1.59 (m, 1 H), 1.58–1.42 (m, 5 H), 1.38 (dd, *J* = 14.5, 7.1 Hz, 1 H), 1.34–1.16 (m, 14 H), 1.00 (d, *J* = 6.7 Hz, 3 H), 0.88 (t, *J* = 7.0 Hz, 3 H) ppm; ^13^C NMR (101 MHz, CDCl_3_) *δ* 138.3, 128.3, 127.6, 127.5, 101.3, 75.6, 74.3, 73.3, 66.8, 66.8, 43.7, 43.2, 37.5, 31.9, 30.3, 29.6, 29.6, 29.3, 25.8, 24.0, 23.6, 22.7, 22.6, 14.1 ppm; HRMS (ESI-TOF): calcd. for C_26_H_44_O_4_ [M + Na]^+^ 443.3137; found 443.3120.

Spectroscopic analysis of (2*R*,4*S*)-**22**: R_f_ = 0.32 (pentane/diethyl ether, 1:1); [α]_D_^20^ = + 3.2 (c = 0.55, CHCl_3_); IR (neat): ν_max_ = 3452, 2960, 2852, 1454, 1263, 1109 cm^−1^; ^1^H NMR (500 MHz, CDCl_3_) δ 7.37–7.22 (m, 5 H), 4.60–4.44 (m, 3 H), 4.06 (dd, *J* = 12.1, 4.2 Hz, 2 H), 3.71 (td, *J* = 12.1, 2.4 Hz, 2 H), 3.32 (d, *J* = 8.9 Hz, 1 H), 3.28 (d, *J* = 8.9 Hz, 1 H), 2.47 (s, 1 H), 2.13–1.95 (m, 1 H), 1.85 (dt, *J* = 13.4, 6.7, 4.6 Hz, 1 H), 1.69–1.56 (m, 1 H), 1.55–1.40 (m, 5 H), 1.36 (dd, *J* = 14.5, 7.0 Hz, 1 H), 1.33–1.14 (m, 14 H), 0.98 (d, *J* = 6.7 Hz, 3 H), 0.87 (t, *J* = 6.8 Hz, 3 H) ppm; ^13^C NMR (101 MHz, CDCl_3_) *δ* 138.3, 128.3, 127.5, 127.5, 101.4, 75.8, 74.3, 73.3, 66.8, 66.8, 43.5, 43.4, 37.4, 31.8, 30.3, 29.6, 29.5, 29.3, 25.8, 24.0, 23.6, 22.7, 22.6, 14.1 ppm; HRMS (ESI-TOF): calcd. for C_26_H_44_O_4_ [M + Na]^+^ 443.3137; found 443.3125.

### 4.8. (2S,4R)-2-((Benzyloxy)methyl)-6-methoxy-4-methyl-2-nonyltetrahydro-2H-pyran (**23**/**24**)

Dioxane (2*R*,4*S*)**-22** (0.230 g, 0.547 mmol) and ZrCl_4_ (0.013 g, 0.055 mmol) was dissolved in anhydrous methanol (0.6 mL) in a 10 mL microwave vial containing a stirrer bar and stirred under microwave irradiation at 50 °C at 100 W for 6 min. The crude product was purified directly by silica gel column chromatography (pentane/diethyl ether, 9:1) to yield **23** and **24** as an inseparable mixture of colourless oils (0.184 g, 89%). (see Appendix A for ^1^H and ^13^C NMR spectra of compounds **23/24**).

Spectroscopic analysis carried out on pure mixture **23/24**: R_f_ = 0.28 (pentane/diethyl ether, 9:1); [α]_D_^20^ = −31.9 (c = 1.0, CHCl_3_); IR (neat): ν_max_ = 2929, 2854, 1454, 1101, 1053 cm^−1^; ^1^H NMR (400 MHz, CDCl_3_) *δ* 7.37–7.21 (m, 5 H), 4.79 (d, *J* = 3.5 Hz, 1 H), 4.61–4.52 (m, 3 H), 4.49 (dd, *J* = 9.8, 2.3 Hz, 1 H), 3.48–3.37 (m, 4 H), 3.31–3.22 (m, 1 H), 2.12–1.94 (m, 1 H), 1.93–1.40 (m, 5 H), 1.37–0.80 (m, 25 H) ppm; ^13^C NMR (101 MHz, CDCl_3_) *δ* 138.8, 138.8, 128.4, 128.4, 127.8, 127.6, 127.6, 99.9, 97.9, 77.4, 77.3, 76.5, 76.0, 73.6, 56.0, 55.7, 40.5, 39.9, 39.1, 39.0, 35.3, 32.1, 30.9, 30.6, 30.4, 29.8, 29.8, 29.5, 25.2, 24.6, 22.8, 22.6, 22.3, 19.9, 14.3 ppm; HRMS (ESI-TOF): calcd. for C_24_H_40_O_3_Na [M + Na]^+^ 399.2875; found 399.2865. 

### 4.9. (2S,4S)-2-((Benzyloxy)methyl)-6-methoxy-4-methyl-2-nonyltetrahydro-2H-pyran (**19**/**20**)

Dioxane (2*S*,4*S*)-**18** (0.291 g, 0.691 mmol) was subjected to the same procedure as **22**. The crude product was purified directly by silica gel column chromatography (pentane/diethyl ether, 9:1) to yield **19** and **20** as an inseparable mixture of colourless oils (0.240 g, 92%). (see Appendix A for ^1^H and ^13^C NMR spectra of compounds **19/20**).

Spectroscopic analysis carried out on pure mixture **19/20**: R_f_ = 0.75 (pentane/ethyl acetate, 9:1); [α]_D_^20^ = −18.3 (c = 0.5, CHCl_3_); IR (neat): ν_max_ = 2923, 2853, 1454, 1376 cm^−1^; ^1^H NMR (400 MHz, CDCl_3_) δ 7.38–7.27 (m, 5 H), 4.73 (d, J = 3.4 Hz, 1 H), 4.58–4.51 (m, 3 H), 3.69 (d, J = 9.0 Hz, 1 H), 3.56–3.40 (m, 3 H), 3.36 (s, 3 H), 1.93 (m, 1 H), 1.83–1.47 (m, 5 H), 1.39–1.14 (m, 19 H), 1.04–0.81 (m, 9 H) ppm; ^13^C NMR (101 MHz, CDCl_3_) δ 138.9, 138.7, 128.5, 128.4, 127.7, 127.6, 127.6, 127.5, 99.7, 98.5, 77.0, 76.3, 73.5, 73.3, 72.7, 70.4, 55.9, 55.5, 40.1, 39.9, 39.6, 39.4, 39.4, 38.8, 32.1, 30.4, 30.4, 29.9, 29.8, 29.8, 29.8, 29.5, 25.5, 23.0, 23.0, 22.9, 22.5, 22.4, 20.3, 14.3 ppm; HRMS (ESI-TOF): calcd. for C_24_H_40_O_3_Na [M + Na]^+^ 399.2875; found 399.2864.

### 4.10. (4R,6S)-6-((Benzyloxy)methyl)-4-methyl-6-nonyltetrahydro-2H-pyran-2-one (**25**)

Acetals **23**/**24** (0.164 g, 0.436 mmol) were dissolved in CH_2_Cl_2_ (13 mL) in a dry 50 mL Schlenk tube containing a magnetic stirrer bar and cooled to 0 °C. *m*-CPBA (0.113 g, <77%, 0.653 mmol) was added followed by BF_3_·OEt_2_ (0.070 mL, 0.566 mmol) and the reaction mixture was stirred at room temperature for 30 min. The reaction mixture was cooled back to 0 °C, quenched slowly with Et_3_N (0.30 mL, 2.18 mmol) and stirred for 30 min. Excess solvent removed in vacuo. The crude product residue was purified by silica gel column chromatography (pentane/diethyl ether, 4:1) to yield **25** as a colourless oil (0.066 g, 42%). (see Appendix A for ^1^H and ^13^C NMR spectra).

Spectroscopic analysis of **25**: R_f_ = 0.36 (pentane/diethyl ether, 3:2); [α]_D_^20^ = −7.0 (c = 0.9, CHCl_3_); IR (neat): ν_max_ = 2929, 2856, 1720, 1454, 1215, 1099 cm^−1^; ^1^H NMR (400 MHz, CDCl_3_) *δ* 7.44–7.22 (m, 5 H), 4.62 (d, *J* = 12.1 Hz, 1 H), 4.54 (d, *J* = 12.1 Hz, 1 H), 3.46 (s, 2 H), 2.62–2.51 (m, 1 H), 2.16–1.97 (m, 2 H), 1.81 (dd, *J* = 13.6, 3.5 Hz, 1 H), 1.74–1.54 (m, 3H), 1.48–1.17 (m, 14 H), 1.04 (d, *J* = 6.0 Hz, 3 H), 0.90 (t, *J* = 6.7 Hz, 3 H) ppm; ^13^C NMR (101 MHz, CDCl_3_) *δ* 171.8, 138.0, 128.4, 127.6, 127.6, 85.1, 75.2, 73.6, 38.2, 37.6, 36.3, 31.8, 30.0, 29.5, 29.4, 29.2, 24.0, 23.3, 22.6, 21.2, 14.1 ppm; HRMS (ESI-TOF): calcd. for C_23_H_36_O_3_Na [M + Na]^+^ 383.2562; 383.2574.

### 4.11. (4.S,6S)-6-((Benzyloxy)methyl)-4-methyl-6-nonyltetrahydro-2H-pyran-2-one (**21**)

Acetals **19**/**20** (0.212 g, 0.563 mmol) were subjected to the same procedure as **23**/**24**. The crude product residue was purified by silica gel column chromatography (pentane/diethyl ether, 4:1) to yield **21** as a colourless oil (0.106 g, 52%). (see Appendix A for ^1^H and ^13^C NMR spectra).

Spectroscopic analysis of **21**: R_f_ = 0.29 (pentane/ethyl acetate, 95:5); [α]_D_^20^ = + 27.75 (c = 0.55, CHCl_3_); IR (neat): ν_max_ = 3017, 2963, 2855, 1717, 1455 cm^−1^; ^1^H NMR (400 MHz, CDCl_3_) *δ* 7.38–7.26 (m, 5 H), 4.56–4.44 (m, 2 H), 3.44 (s, 2 H), 2.61–2.53 (m, 1 H), 2.22–2.08 (m, 1 H), 2.05–1.97 (m, 1 H), 1.88 (dd, J = 17.5, 12.1 Hz, 1 H), 1.72–1.53 (m, 2 H), 1.45–1.18 (m, 15 H), 0.96 (d, J = 6.4 Hz, 3 H), 0.88 (t, J = 6.9 Hz, 3 H) ppm; ^13^C NMR (101 MHz, CDCl_3_) *δ* 171.7, 137.9, 128.6, 127.9, 127.8, 84.7, 74.1, 73.7, 39.3, 38.4, 37.2, 32.0, 30.0, 29.7, 29.5, 23.9, 22.8, 22.8, 21.7, 14.3 ppm; HRMS (ESI-TOF): calcd. for C_23_H_36_O_3_Na [M + Na]^+^ 383.2562; found 383.2558.

### 4.12. (4. R,6S)-4-Methylmalyngolide (**8**)

In a 10mL conical flask containing a magnetic stirrer bar, protected lactone **25** (0.045 g, 0.125 mmol) was dissolved in ethyl acetate (2 mL) and Pd(OH)_2_/C (20 wt.%) (0.0018 g, 0.0125 mmol) was added. The reaction vessel was placed in a Parr reactor under 25 bar H_2_ pressure for 72 h. The reaction was monitored by TLC (pentane/diethyl ether, 1:1). Upon reaction completion, the crude product was run through a small silica gel column (ethyl acetate) to yield (4*R*, 6*S*)-4-methylmalyngolide **8** as a colourless oil (0.022 mg, 65%). (see Appendix A for ^1^H and ^13^C NMR spectra).

Spectroscopic analysis of **8**: R_f_ = 0.08 (pentane/diethyl ether, 1:1); [α]_D_^20^ = −14.8 (c = 0.7, CHCl_3_); IR (neat): ν_max_ = 3423, 2924, 2854, 1722, 1458, 1377, 1246, 1088 cm^−1^; ^1^H NMR (400 MHz, CDCl_3_) δ 3.68 (d, J = 12.0 Hz, 1H), 3.50–3.41 (m, 1H), 2.60 (ddd, J = 17.2, 4.4, 2.2 Hz, 1H), 2.16–2.03 (m, 1H), 1.97 (dd, J = 17.2, 12.0 Hz, 1H), 1.79–1.67 (m, 2H), 1.64–1.51 (m, 2H), 1.42 (s, 1H), 1.37–1.19 (m, 14H), 1.03 (d, J = 6.3 Hz, 3H), 0.87 (t, J = 6.9 Hz, 3H) ppm; ^13^C NMR (101 MHz, CDCl_3_) *δ* 171.6, 86.6, 67.7, 38.1, 36.5, 34.6, 31.8, 30.0, 29.5, 29.4, 29.2, 23.8, 23.5, 22.6, 21.4, 14.1 ppm; HRMS (ESI-TOF): calcd. for C_16_H_30_O_3_Na [M + Na]^+^ 293.2093; found 293.2088.

### 4.13. (4S,6S)-4-Methylmalyngolide (**7**)

Protected lactone **21** (0.075 g, 0.208 mmol) was subjected to the same procedure as **25**. Upon reaction completion, the crude product was run through a small silica gel column (ethyl acetate) to yield (4*S*,6*S*)-4-methylmalyngolide **7** as a colourless oil (0.053 mg, 94%). (see Appendix A for ^1^H and ^13^C NMR spectra).

Spectroscopic analysis of **7**: R_f_ = 0.10 (pentane/diethyl ether, 1:1); [α]_D_^20^ = + 45.3 (c = 0.35, CHCl_3_); IR (neat): ν_max_ = 3018, 2928, 1711, 1215 cm^−1^; ^1^H NMR (400 MHz, CDCl_3_) *δ* 3.64 (d, J = 11.7 Hz, 1 H), 3.58 (d, J = 11.7 Hz, 1 H), 2.59 (ddd, J = 17.4, 4.5, 2.3 Hz, 1 H), 2.29–2.17 (m, 1 H), 1.95–1.84 (m, 2 H), 1.69–1.57 (m, 2 H), 1.43–1.18 (m, 16 H), 0.99 (d, J = 6.3 Hz, 3 H), 0.87 (t, J = 6.9 Hz, 3 H) ppm; ^13^C NMR (101 MHz, CDCl_3_) *δ* 172.1, 86.1, 68.0, 38.4, 38.4, 37.1, 32.0, 30.0, 29.6, 29.4, 24.4, 23.1, 22.8, 21.6, 14.2 ppm; HRMS (ESI-TOF): calcd. for C_16_H_30_O_3_Na [M + Na]^+^ 293.2093; found 293.2080.

## 5. Materials and Methods—Biological Testing

### 5.1. Preparation of Compounds

Samples were reconstituted into an appropriate volume of DMSO to achieve a final concentration of 10 mg/mL.

### 5.2. Antibacterial Activity Testing—Determination of Minimum Inhibitory Concentration (MIC) and Minimum Bactericidal Concentration (MBC)

Samples of each of these chemical compounds were reconstituted into an appropriate volume of DMSO to achieve a final concentration of 10 mg/mL. MIC values for these compounds was determined by two-fold broth microdilution in 96-well microtiter plates. Briefly, overnight cultures of *Escherichia coli* ATCC 25922, *Escherichia coli* 4, MRSA ATCC 43300 and MRSA 06/04 (see Appendix A for further information about the isolates) were diluted in sterilised PBS to approximately 10^5^ CFU/mL. Aliquots of 5 µL were then transferred to separate wells in a 96-well plate that contained 100 µL of each compound at varying concentrations (ranging from 100–0.195 μg/mL) prepared from two-fold serial dilutions in Mueller-Hinton (MH) broth. Plates were incubated at 37 °C for 18 h using an Omnilog^®^ automated incubator (Biolog Inc.;21124 Cabot Boulevard, Hayward, CA 94545, USA) and MIC values recorded.

Determination of the MBC values for all compounds tested above was performed in MH broth media. Again, 5µL were collected from the MICs 96-well plates (above) and re-inoculated into fresh sterile 96-well plates containing fresh MH. Plates were incubated under the same conditions mentioned above. The assay was performed in triplicate for each compound. (see Appendix A for UCD Centre for Food Safety strains used for determination of antibacterial activity and Appendix A for Antibacterial activity of compounds tested – MIC and MBC results (triplicates)).

## Figures and Tables

**Figure 1 ijms-22-06400-f001:**
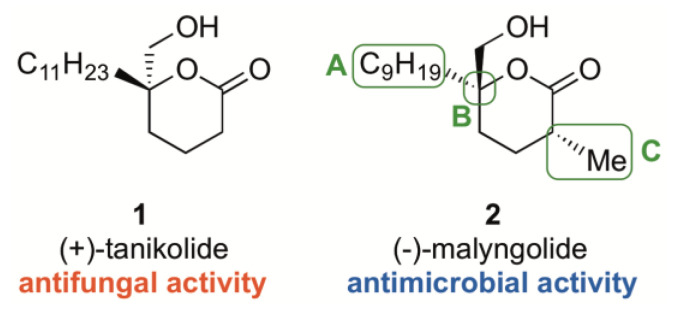
(+)-Tanikolide and (−)-malyngolide.

**Figure 2 ijms-22-06400-f002:**
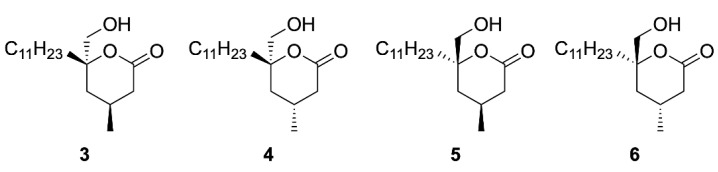
β-Methyl-(+)-tanikolide based analogues **3**–**6**.

**Figure 3 ijms-22-06400-f003:**
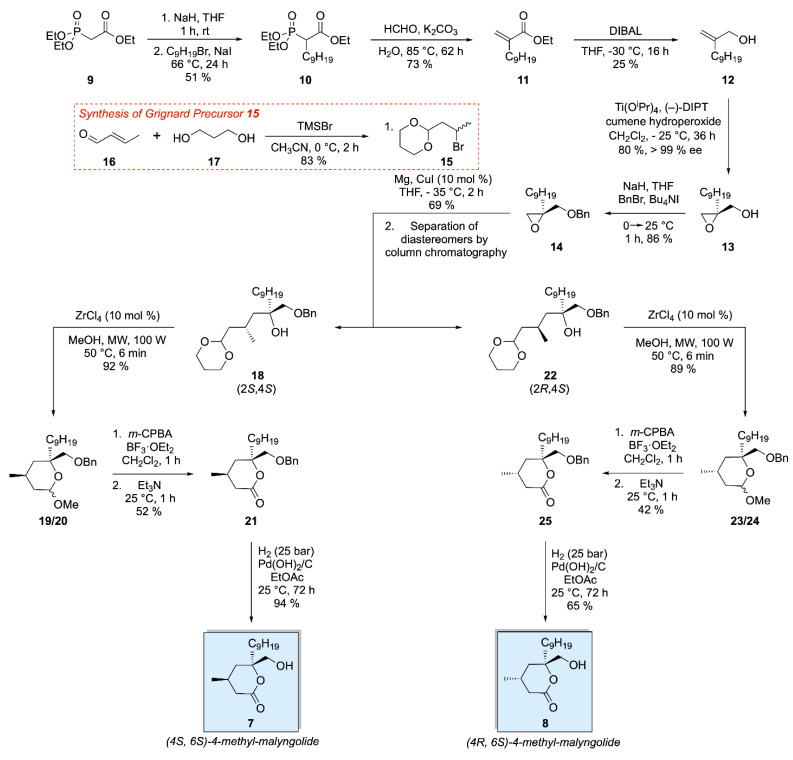
Synthesis of malyngolide analogues **7** and **8**.

**Table 1 ijms-22-06400-t001:** Antibacterial activity of **3**–**6**—MIC and MBC results (triplicates) ^[a]^.

Compound	*E. coli* 25922		*E. coli* 4		MRSA ATCC 43300		MRSA 06/04	
	MIC ^[b]^	MBC	MIC	MBC	MIC	MBC	MIC	MBC
**3**	>100	>100	>100	>100	>100	>100	>100 *	>100
**4**	>100	>100	>100	>100	>100	>100	>100 *	>100
**5**	>100	>100	>100	>100	**12.5**	**12.5**	**12.5**	**50**
**6**	>100	>100	>100	>100	>100	>100	>100 *	>100

^[a]^ * denotes a change in strain phenotype ^[b]^ MIC—minimum inhibitory concentration, MBC—minimum bactericidal concentration. Values are given in μg/mL. Bold-face values denote compounds that showed activity against the tested bacteria. The maximum concentration of compound tested in each case was 100 μg/mL.

**Table 2 ijms-22-06400-t002:** The MIC and MBC measurements for compounds **7** and **8**.

Compound	*E. coli* 25922		*E. coli* 4		MRSA ATCC 43300		MRSA 06/04	
	MIC ^[a]^	MBC	MIC	MBC	MIC	MBC	MIC	MBC
**7**	>100	>100	>100	>100	**50**	**50**	**50**	**50**
**8**	>100	>100	>100	>100	**50**	**100**	**50**	**50**

^[a]^ MIC—minimum inhibitory concentration, MBC—minimum bactericidal concentration. Values are given in μg/mL. Bold-face values denote compounds that showed activity against the tested bacteria. The maximum concentration of compound tested in each case was 100 μg/mL.

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
