# Peer review of "Investigation of the Anti-Methicillin-Resistant Staphylococcus aureus Activity of (+)-Tanikolide- and (+)-Malyngolide-Based Analogues Prepared by Asymmetric Synthesis"

_ijms, 2021, doi:10.3390/ijms22126400_

Round 1

Reviewer 1 Report

The Authors report some studies about the antibacterial activity of some organic compounds analogues to the known (+)-Tanikolide- and (+)-Malyngolide.

Although the paper contains some interesting data, in order to publish them would be necessary to properly describe the presented study and the results obtained inside the opportune context. The current form of the manuscript shows some severe issues, as:

- Abstract: very short and not explicative.

- Introduction: just the results obtained by the Authors are reported, without any kind of introduction and description of the general context, as well as the reporting of similar studies by other Authors. Why this class of molecules is important? what has been done previously? in which extend the presented research fits in the existing literature and improves it?

- Discussion: again, a proper comparison with the literature it is not provided. In addition, for the reader is impossible to understand to which compounds the Authors are referring too, because they are not described or named in the first part of the paragraph.

e.g., the discussion starts in this way: “The four β-methyl analogues and a number of intermediates were submitted for biological testing to ascertain if they had any antifungal and antimicrobial activity.” The reader should know exactly which analogues and how many intermediates and their exact structure.

Minor issue: very low quality of the figure.

I suggest a deep rewriting of the manuscript prior to a further submission.

Reviewer 2 Report

The research work done by Patrick J. Guiry research group “Investigation of the Anti-Methicillin-Resistant Staphylococcus aureus Activity of (+)-Tanikolide- and (+)-Malyngolide-Based Analogues Prepared by Asymmetric Synthesis” describes a good protocol for the synthesis of Tanikolide  and    Malyngolide analogues. Authors documented synthesis of these molecules which are having antibacterial activity towards MRSA. The major advantage of the present documented report is the synthesized molecules having activity towards gram positive bacteria which makes the report attractive for readers. There are not many literature procedures, similar to present documented molecules in the literature for the synthesis of this particular compounds and also study towards antibacterial activity. These products are important in organic synthesis and medicinal chemistry development and this antibacterial study has broad applicability towards development of antibacterial scaffolds for further application and development. Given the importance of practicality for this work, I recommend the publication of this manuscript in the international journal of molecular sciences.

Author Response

There are no comments to address from this reviewer.

Round 2

Reviewer 1 Report

The revised version of the manuscript provided by the Authors results highly improved with respect to the first one. In the current form it is possible to easy follow the analysis reported and to understand the value of the results reported. 
Some minor suggestions.

- Please substitute numbers with names in the abstract.

- Define MIC in the abstract. It is defined in the Table 1 of the manuscript but it should be defined also here.

- Line 58: @-methyl

- Figure 2: the quality of the picture is still very low

- Line 76: @-  ?

- Line 89: please, correct “will be tested”

- Line 159: 1H NMR
